# Awareness of antimicrobial resistance and antibiotic use among poultry farmers in Accra, Ghana: A cross-sectional survey

Henry Kwadwo Hackman[1]*, Lawrence Annison[1], Reuben Essel Arhin[2], Ebenezer Krampah Aidoo[1], Evans Andoh Wilberforce[1], Alice Constance Mensah[3], Louis Appiah[3], Sharon Annison[4], Matilda Ayim-Akonor[5]

**1** Department of Medical Laboratory Technology, Faculty of Applied Sciences, Accra Technical University, Accra, Ghana, **2** Department of Science Laboratory Technology, Faculty of Applied Sciences, Accra Technical University, Accra, Ghana, **3** Department of Applied Mathematics and Statistics, Faculty of Applied Sciences, Accra Technical University, Accra, Ghana, **4** Department of Epidemiology and Disease Control, School of Public Health, University of Ghana, Accra, Ghana, **5** Animal Health Division, Council for Scientific and Industrial Research (CSIR)-Animal Research Institute, Accra, Ghana

* hkhackman@atu.edu.gh

## Abstract

### Background

Antimicrobial resistance (AMR) in livestock production is an escalating global public health threat, fuelled in part by the misuse of antibiotics in animal farming. This study aimed to assess the knowledge, attitudes, and practices regarding antimicrobial use and AMR among poultry farmers in Accra, Ghana.

### Methods

A cross-sectional survey of 400 poultry farmers in five communities across the Greater Accra Region was conducted using a structured questionnaire. The survey collected data on demographics, AMR awareness, antibiotic usage practices (treatment, prophylaxis, or growth promotion), and commonly used antibiotic classes from 14th March to 26th September 2023. Descriptive statistics summarized the findings. Associations between farmer characteristics (age, gender, education, location) and antimicrobial self-medication (treating poultry without veterinary consultation) were evaluated using chi-square tests. A multivariate logistic regression model identified independent demographic predictors of self-medication, with significance set at $p < 0.05$.

### Results

Most farmers (70%) reported self-medicating sick poultry without veterinary consultation, and only 35% of respondents were aware of the concept of AMR. Antibiotics were predominantly used for treating illness (63% of farmers), while 20% used them

**Data availability statement:** All relevant data are within the paper and its Supporting Information files.

**Funding:** The author(s) received no specific funding for this work.

**Competing interests:** The authors have declared that no competing interests exist.

for prophylaxis and only 4% for growth promotion. The most administered antibiotic classes were tetracyclines (26%), nitrofurans (24%), aminoglycosides (18%), penicillins (17%), and fluoroquinolones (10%). Male farmers and those over 30 years had significantly higher rates of antimicrobial self-medication than female and younger farmers (p < 0.001), and in multivariate analysis, being male (odds ratio ~4.9), age > 30 years (OR ~4.6), and farming in a rural area (OR ~2.7) were independent predictors of self-medication with antibiotics.

## Conclusions

Inappropriate antibiotic use is highly prevalent among poultry farmers in Accra, and awareness of AMR is low. These findings underscore an urgent need to strengthen veterinary oversight and enforce regulations on antibiotic sales. Educating farmers through a One Health approach is also recommended to promote prudent antibiotic use and curb the rise of antimicrobial resistance.

---

## Introduction

The poultry industry plays an important role in Ghana's economy by providing employment and contributing to the gross domestic produce [1]. Poultry production has grown significantly in West Africa in recent decades to meet increasing demand for animal protein [2]. This rapid expansion, in Ghana and globally, has been accompanied by extensive use of antimicrobials in poultry husbandry for disease prevention and growth promotion [3]. Global antimicrobial use in food animals is projected to increase by 67% from 2010 to 2030, with especially sharp rises in low- and middle-income countries such as Ghana [1,3]. In many large-scale farms, antibiotics are routinely administered, sometimes without adequate veterinary oversight [4–5]. Improper or excessive antibiotic use in animals can select for antibiotic-resistant bacteria and lead to drug residues in animal products [1,3]. These resistant pathogens and antibiotic residues can be transmitted to humans through the food chain or the environment, contributing to the AMR crisis in Ghana [3].

In poultry farming, farmers may misuse antibiotics by administering them without professional guidance, using incorrect dosages, or not observing withdrawal periods before slaughter [6]. Such practices facilitate the emergence and spread of antimicrobial-resistant bacteria and compromise the effectiveness of essential antibiotics. There is limited information on the awareness of the antimicrobial resistances and antibiotic usage among poultry farmers in Accra, Ghana.

This study was conducted to assess the knowledge, attitudes, and practices (KAP) of poultry farmers in selected areas of Accra, Ghana, concerning antimicrobial use and resistance, and to identify the most used antibiotics in their operations. By identifying gaps in awareness and risky usage practices, the findings can inform targeted education and policy measures to mitigate AMR development in the poultry sector.

## Materials and methods

### Study design and setting

A cross-sectional survey among poultry farmers was conducted in five communities of the Greater Accra Region of Ghana: Nima, Awoshie, Mallam Atta, Pokuase, and Ayikuma. The survey collected data on demographics, AMR awareness, antibiotic usage practices (treatment, prophylaxis, or growth promotion), and commonly used antibiotic classes from 14th March to 26th September 2023. These areas include both urban and rural settings with active poultry farming. Nima (Ayawaso East Municipality) is a densely populated urban area (population ~53,000) known for its vibrant markets [7]. Awoshie (Ga Central/Ablekuma North) is a bustling suburb with many residents engaged in trading and small businesses. Mallam Atta, near Accra New Town, hosts a major livestock market. Pokuase (Ga West Municipality) is a rapidly developing rural community. Ayikuma (Shai Osudoku District) is a rural township. This mix of urban and rural communities allowed comparison of farming practices across different settings.

### Sample size and sampling

Using Cochran's formula for cross-sectional studies

$$n = \frac{Z^2 \times P(1-P)}{e^2},$$

where:

 n = sample size (400)

 Z = abscissa of the normal curve that cuts off an area (α) at the tail (1.96)

 e = the acceptable sample error at 95% confidence interval (0.05).

 P = the estimated proportion of the attribute present in the population (0.5).

 A minimum sample of ~384 was calculated. The study targeted 400 farmers to enhance representativeness and statistical power.

 Participants were selected from farmer lists obtained from farmer associations and agriculture extension offices and by visiting local farms and markets. A systematic sampling technique was then used to randomly select participants from these lists. All 400 selected farmers agreed to participate, resulting in a 100% response rate. Eligible respondents were poultry farmers (backyard or commercial) aged 18 or above who had been raising poultry for at least one year. In each community, researchers approached farmers at poultry farms or selling points and invited them to participate. After obtaining written or verbal informed consent, a structured questionnaire was administered in the respondent's preferred language (English or a local language).

### Data collection

The questionnaire (S1 File) was administered to respondents to complete and those who were illiterate were interviewed in the local language to respond to the questions. The questionnaire collected information on: (1) Demographics – age, gender, education level, and farm location (urban vs rural); (2) Knowledge and Awareness – familiarity with the term "antimicrobial use (AMU)," understanding of AMR (e.g., whether using antimicrobials in animal production can lead to resistant infections, whether avoiding overuse can prevent AMR); (3) Practices – how farmers obtain and use antimicrobials (sources of drugs, whether they self-treat animals or consult veterinarians, adherence to recommended guidelines or dosages, purposes for which antibiotics are used such as treatment, prevention, or growth promotion); (4) Attitudes – perceptions about prudent antibiotic use (e.g., importance of veterinary advice, belief in alternatives like vaccines and hygiene, support for regulations on antibiotic use, etc.); and (5) Commonly Used Antibiotics – which antibiotic classes the farmer most frequently administers to their poultry. The questionnaire mostly used closed-ended questions (yes/no or

multiple-choice, including Likert-scale items for attitudes). It was developed in English and reviewed by experts for clarity and relevance, then pre-tested with 10 poultry farmers outside the study areas, leading to minor wording revisions for understanding. Farmers who could not accurately recall the names or classes of antibiotics they used were shown pictures of common drug packages and asked to identify what they used on their farms. Farmers who did not send their sick animals to the veterinary clinic for treatment were deemed to be practicing antimicrobial self-medication (ASM). Positive attitudes refer to farmers seeking advice and supporting regulations to control antimicrobial resistance development.

### Ethical considerations

The protocol used for data collection was approved by the Ethical Review Committee of Accra Technical University with protocol identification number ATU/MLT/ET/002/2022–2023. The objectives of the study, risks, benefits, right to refuse and confidentiality were explained to the respondents prior to obtaining written informed consent (S1 File) to voluntarily participate in the survey. The identity and information on the respondents were not disclosed. The authors do not have access to information that could identify individual participants during or after data collection (S2 Data).

### Data analysis

Completed questionnaires were coded and entered into SPSS (Version 25) for analysis. Descriptive statistics (frequencies and percentages) were computed for all survey items. Key results are presented in charts for clearer visualization.

Chi-square Tests: Chi-square tests were used to assess associations between socio-demographic factors (independent variables: age group, gender, education, and location) and the main outcome variable of antimicrobial self-medication (ASM) – defined as farmers administering antibiotics to their poultry without veterinary consultation. A significance level of $p < 0.05$ was considered statistically significant for these bivariate tests. A Bonferroni adjustment was applied to the significance threshold to minimize the risk of inflated *Type I error* due to multiple chi-square tests.

Multivariant Analysis: Variables that showed significant associations in chi-square analyses were then included in a multivariable logistic regression model to identify independent predictors of antimicrobial self-medication. The logistic regression results are reported as odds ratios (OR) with 95% confidence intervals (CI) and p-values. All statistical tests were two-tailed.

Model Diagnostics: The Hosmer–Lemeshow goodness-of-fit test (S3 Table) was used to evaluate logistic model calibration. A non-significant result ($p > 0.05$) indicated acceptable model fit. Multicollinearity was assessed using variance inflation factors (VIFs), with values $< 5$ considered acceptable. Model discrimination was further assessed using the area under the Receiver Operating Characteristic (ROC) curve (AUC).

## Results

### Socio-demographic characteristics of poultry farmers

As indicated in Table 1, a total of 400 poultry farmers participated in the survey, of whom 248 (62.0%) were male and 152 (38.0%) were female. The respondents' ages ranged from 18 to 65 years, with a majority (340 farmers, 85%) above 30 years old. About 15% of participants were between 18 and 30 years, 43% were 31–40 years, and 42% were over 40. Regarding education, 152 farmers (38.0%) had no formal education and were considered illiterate. Among those with formal schooling, 100 (25.0%) had primary-level education, another 100 (25.0%) had secondary education, and only 48 (12.0%) had attained tertiary (post-secondary) education. The participants were evenly split by location: 200 farmers (50%) were from urban communities and 200 (50%) from rural communities.

### Antimicrobial resistance awareness and knowledge

Knowledge of antimicrobial use and resistance was generally low. Only 140 farmers (35.0%) had heard of the term "antimicrobial use" (AMU) or understood what it meant, while the majority (260 farmers, 65.0%) had no familiarity with

**Table 1. Socio-demographic characteristics of respondents (n = 400).**

| Variable | Category | Count | Frequency (%) |
|---|---|---|---|
| **Gender** | Male | 248 | 62.0% |
| | Female | 152 | 38.0% |
| | Total | 400 | 100.0% |
| **Age** | 18–30 | 60 | 15.0% |
| | 31–40 | 172 | 43.0% |
| | > 40 | 168 | 42.0% |
| **Education** | Illiterate | 152 | 38.0% |
| | Primary | 100 | 25.0% |
| | Secondary | 100 | 25.0% |
| | Tertiary | 48 | 12.0% |
| **Geographical Location** | Urban | 200 | 50.0% |
| | Rural | 200 | 50.0% |

the term. When asked if using antimicrobials in animal production could boost the rate of AMR development, 172 (43.0%) answered "Yes," acknowledging this risk, but 140 (35.0%) said "No," and 88 (22.0%) responded that they *did not know*. Similarly, awareness of prevention measures was limited: only 140 farmers (35.0%) agreed that avoiding the overuse of antimicrobials in poultry could reduce the development of resistance, whereas 152 (38.0%) did not believe overuse avoidance would make a difference, and 108 (27.0%) were unsure. On the question of whether imprudent use of antimicrobials could negatively affect the health of other people, just over half – 208 respondents (52.0%) – answered "Yes." However, 80 farmers (20.0%) did not think that misuse of antibiotics in their farm could impact others' health, and 112 (28.0%) had no idea that their antibiotic practices might contribute to risks beyond their farm. These findings as shown in Table 2 indicate significant gaps in awareness.

## Antimicrobial usage practices

The survey in Table 3 revealed widespread antibiotic self-medication in poultry management. The results indicated that 132 farmers (33.0%) treated the birds themselves with medications on hand, and an additional 148 (37.0%) indicated they consult other experienced farmers for advice before treating the birds. Hence 280 (70.0%) farmers effectively practiced antimicrobial self-medication. Only 120 farmers (30.0%) reported that they take their sick poultry to a veterinarian for diagnosis and treatment, while none of the respondents chose to do nothing. In line with this, when it comes to who administers antimicrobials to the birds, 214 farmers (53.5%) said they personally give the drugs to their poultry, compared to 160 (40.0%) who have a veterinarian administer the medications. A small fraction (26 farmers, 6.5%) relies on local traditional healers or informal animal health assistants to treat their birds.

Most farmers also do not adhere to formal guidelines for antibiotic use. A large majority (280 farmers, 70.0%) do not refer to any standard guidelines or dosage instructions when administering antimicrobials, instead using their own experience or advice from drug sellers. Only 120 (30.0%) indicated that they consulted guidelines or product literature on proper antibiotic use. Additionally, while veterinarians are not always consulted for treatment, many farmers do obtain some advice at the point of purchase: 268 (67.0%) said they receive a prescription or recommendation from veterinarians before buying antibiotics (often through agro-veterinary shops), whereas 132 (33.0%) admitted they purchase and use antibiotics without any prescription.

Regarding the purposes for which antimicrobials are used, the majority (252 farmers, 63.0%) use antibiotics primarily for treating sick poultry (therapeutic use). However, a substantial subset reported non-therapeutic uses: 80

**Table 2. Knowledge of antimicrobial usage and antimicrobial resistance in poultry production (n = 400).**

| Awareness of Antimicrobial usage and AMR | Category | Count | Frequency (%) |
|---|---|---|---|
| Knowledge of Antimicrobial Use | Yes | 140 | 35.0% |
| | No | 260 | 65.0% |
| | Total | 400 | 100.0% |
| Knowledge of the use of antimicrobials in animal production boosts the rate of AMR development. | Yes | 172 | 43.0% |
| | No | 140 | 35.0% |
| | I don't know | 88 | 22.0% |
| Avoiding overuse of antimicrobials in animal production can reduce AMR development. | Yes | 140 | 35.0% |
| | No | 152 | 38.0% |
| | I don't know | 108 | 27.0% |
| Imprudent use of antimicrobials affect the health of others in the form of AMR. | Yes | 208 | 52.0% |
| | No | 80 | 20.0% |
| | I don't know | 112 | 28.0% |

**Table 3. Practice of antimicrobial usage & antimicrobial resistance in poultry production (n = 400).**

| Practice of Antimicrobial Usage | Category | Count | Frequency (%) |
|---|---|---|---|
| **What do you do when your animals get sick?** | Self-treat | 132 | 33 |
| | Take to Veterinary | 120 | 30 |
| | Consult other farmers | 148 | 37 |
| | Nothing | 0 | 0 |
| **Who administers antimicrobials for your birds?** | Self-administration | 214 | 53.5 |
| | Veterinarian | 160 | 40 |
| | Local traditional healer | 26 | 6.5 |
| **Did you refer to guidelines while you administer antimicrobials for your animals?** | Yes | 120 | 30 |
| | No | 280 | 70 |
| **Did you get prescription from veterinarians before you buy drugs?** | Yes | 268 | 67 |
| | No | 132 | 33 |
| **For what purpose did you use antimicrobials most?** | Treatment | 252 | 63 |
| | Control (metaphyl) | 52 | 13 |
| | Prevention (prophylaxis) | 80 | 20 |
| | Increase in production | 16 | 4 |
| **Source of antimicrobials for your animals?** | Local Dispensers | 152 | 38 |
| | Veterinary Clinic | 120 | 30 |
| | Veterinary Pharmacy | 128 | 32 |

farmers (20.0%) mainly use antibiotics prophylactically for disease prevention, administering them to healthy birds to ward off potential infections. Another 52 farmers (13.0%) mostly use antibiotics for metaphylaxis (control), meaning they treat an entire flock when some birds show signs of disease, to control spread. Alarmingly, 16 farmers (4.0%) said the primary reason they give antimicrobials is to promote growth or increase production.

Farmers reported obtaining their antimicrobials from various sources. The most common sources were local agro-chemical shops or dispensers, cited by 152 farmers (38.0%). These are retail shops in the community that sell veterinary drugs over the counter. Veterinary pharmacies were the source for 120 farmers (30.0%), and veterinary clinics for 128 farmers (32.0%). [24].

## Attitudes toward antimicrobial use and resistance

The farmers' attitudes toward antimicrobial stewardship were assessed through several statements. The responses reveal generally positive attitudes in principle, albeit with a notable proportion of uncertainty (neutral or "don't know") for many questions as indicated in Table 4.

For the statement "Professional advice should be sought before using antimicrobials", a strong majority concurred: 172 farmers (43.0%) *strongly agreed* and 132 (33.0%) *agreed* that consulting a veterinarian or professional is recommended before antibiotic use, totalling 76% in agreement. Only 8 farmers (2.0%) disagreed, none strongly disagreed, and 72 (18.0%) were neutral on this point, while 16 (4.0%) said they had no idea.

When asked if "imprudent use of antimicrobials can result in an irreversible loss of drug effectiveness", reflecting understanding of resistance development, 112 (28.0%) strongly agreed and 36 (9.0%) agreed (total ~37% agreeing). However, a significant number (120 farmers, 30.0%) chose a neutral stance, indicating uncertainty, and 52 (13.0%) disagreed with the statement. Additionally, 28 (7.0%) strongly disagreed that careless use would impact drug efficacy, and 52 (13.0%) admitted they *don't know*.

Regarding the role of alternatives to antibiotics, attitudes were again split. To the statement "Using antimicrobial alternatives like good hygienic practices and vaccination could reduce AMR development", 72 farmers (18.0%) strongly agreed and another 72 (18.0%) agreed (36% agreeing). Meanwhile, 80 (20.0%) were neutral, 36 (9.0%) disagreed, 8 (2.0%) strongly disagreed, and a large proportion – 132 farmers (33.0%) – responded that they *don't know*.

On the issue of antibiotics used for productivity, farmers think "using antimicrobials for the purpose of animal production (growth promotion) is abusing the drugs.**"** Here, opinions varied: 72 (18.0%) strongly agreed and 120 (30.0%) agreed (48% in agreement) that administering antibiotics for growth promotion constitutes misuse. In contrast, 72 (18.0%) were neutral, and another 72 (18.0%) disagreed (i.e., they do *not* think it's abuse). A small number, 16 (4.0%), strongly disagreed that it is abuse, and 8 (2.0%) didn't know.

There was stronger agreement regarding regulatory measures. For the statement "Antimicrobial use regulations could be a solution for the irrational use of antimicrobials in animal production", more than half of respondents supported it: 72 (18.0%) strongly agreed and 152 (38.0%) agreed (total 56% agreement). 80 (20.0%) remained neutral on this issue, 72 (18.0%) disagreed, 8 (2.0%) strongly disagreed, and 16 (4.0%) had no opinion.

Finally, on public awareness, farmers largely believed that education could help: "Public awareness can reduce the development of AMR" drew 212 respondents (53.0%) who agreed or strongly agreed in total. Specifically, a substantial

**Table 4.  Attitude of antimicrobial usage & antimicrobial resistance in poultry production.**

| Questions/ Responses (Frequency/ %) | 1 | 2 | 3 | 4 | 5 | 6 |
|---|---|---|---|---|---|---|
| Is professional advice before using antimicrobials recommended? | 172(43) | 132(33) | 72(18) | 8(2) | 0 | 16(4) |
| Can imprudent AMU result irreversible loss of drugs effectiveness? | 112(28) | 36(9) | 120(30) | 52(13) | 28(7) | 52(13) |
| Can using antimicrobial alternatives like, good hygienic practice and vaccination reduce AMR development | 72(18) | 72(18) | 80(20) | 36(9) | 8(2) | 132(33) |
| Do you think using antimicrobials for the purpose of animal production is abusing antimicrobials? | 72(18) | 120(30) | 112(28) | 72(18) | 16(4) | 8(2) |
| Can AMU regulations be a solution for the irrational use of antimicrobials in animal production? | 72(18) | 152(38) | 80(20) | 72(18) | 8(2) | 16(4) |
| Can public awareness create a reduction in the development of AMR? | 280(20) | 132(33) | 80(20) | 52(13) | 28(7) | 28(7) |

*Key: 1 = Strongly agree; 2 = Agree; 3 = Neutral; 4 = Disagree; 5 = Strongly disagree; 6 = I don't know.*

280 farmers (70.0%) strongly agreed or expressed agreement in some form, and an additional 132 (33.0%) agreed – indicating broad consensus that outreach and education would be beneficial. Only 80 participants (20.0%) in total disagreed with this idea, and 28 (7.0%) were unsure.

## Commonly used antibiotics in poultry farming

All respondents were identified as using antibiotics in their animals. Farmers were asked which antibiotic classes they most frequently use in their poultry. The results (each farmer naming the one they use most often) showed that a wide range of antibiotic classes are being employed on poultry farms, many of which are classified as critically or highly important in human medicine. Tetracyclines were the most used antibiotics, cited by 104 farmers (26.0%). Tetracyclines (such as oxytetracycline) are broad-spectrum agents often available as poultry formulations and were frequently self-administered by farmers to treat suspected bacterial infections like respiratory or enteric illnesses. The second most common were nitrofurans, reported by 96 farmers (24.0%).

Following these, aminoglycosides (such as streptomycin, neomycin, or gentamicin) were the main antibiotics for 72 farmers (18.0%). Penicillins (e.g., amoxicillin or ampicillin) were next, used by 68 farmers (17.0%). Together, these four classes (tetracyclines, nitrofurans, aminoglycosides, penicillins) accounted for the primary antibiotics used by most farmers. Another 40 farmers (10.0%) primarily used fluoroquinolones (such as enrofloxacin or ciprofloxacin) – a class of high importance in human medicine often used to treat serious poultry infections but whose overuse can quickly select for resistant bacteria. Less commonly used classes included macrolides by 10 farmers (2.5%), sulfonamides/trimethoprim (co-trimoxazole) by 6 farmers (1.5%), and polymyxins (colistin) by 4 farmers (1.0%) as indicated in Fig 1.

## Factors associated with antimicrobial self-medication

The study explored how farmers' demographic factors relate to their likelihood of antimicrobial self-medication (ASM) (treating poultry with antibiotics without veterinary guidance). Table 5 and 6 summarizes the associations. In bivariate analyses (chi-square tests), gender, age, education, and geographic location were all significantly associated with self-medication practices ($p < 0.001$ for each). After applying a Bonferroni correction ($\alpha = 0.0125$ [0.05/4]), gender, age, and location remained significant, while education lost significance as indicated in Table 5 and 6.

The results indicated that 70% of the poultry farmers practiced antimicrobial self-medication by not sending their sick animals to the veterinary clinic for treatment as indicated in Table 5. Male farmers were far more likely to engage in ASM

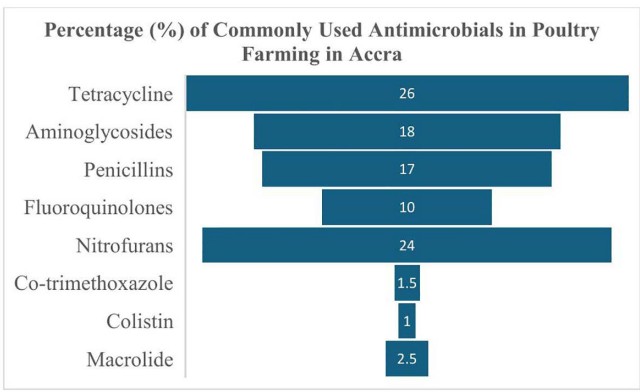

**Fig 1. Commonly used antimicrobial by poultry farmers in Accra.**

**Table 5. Frequency and percentage of antimicrobial self-medication (ASM) among poultry farmers by socio-demographic characteristics (N = 400).**

| Factor | Category | No ASM, n (%) | ASM, n (%) | Total |
|---|---|---|---|---|
| **Gender** | Female (n=152)<br>Male (n=248)<br>Total | 77 (50.7)<br>43 (17.3)<br>120 (30.0) | 75 (49.3)<br>205 (82.7)<br>280 (70.0) | 152 (100.0)<br>248 (100.0)<br>400 (100.0) |
| **Age** | ≤30 yrs (n=60) | 36 (60.0) | 24 (40.0) | 60 (100.0) |
|  | >30 yrs (n=340) | 84 (24.7) | 256 (75.3) | 340 (100.0) |
| **Education** | Illiterate (n=152) | 40 (26.0) | 112 (74.0) | 152 (100.0) |
|  | Primary (n=100) | 12 (12.0) | 88 (88.0) | 100 (100.0) |
|  | Secondary (n=100) | 32 (32.0) | 68 (68.0) | 100 (100.0) |
|  | Tertiary (n=48) | 36 (75.0) | 12 (25.0) | 48 (100.0) |
| **Location** | Urban (n=200) | 80 (40.0) | 120 (60.0) | 200 (100.0) |
|  | Rural (n=200) | 40 (20.0) | 160 (80.0) | 200 (100.0) |

**Table 6. Chi-square analysis of socio-demographic variables and antimicrobial self-medication (ASM).**

| Variable | ASM (%) | $\chi^2$ (df) | *p*-value | Significance (Bonferroni α=0.0125) |
|---|---|---|---|---|
| Gender | 82.7 (male) vs 49.3 (female) | 45.6 (1) | <0.001 | Significant |
| Age | 75.3 (>30 yrs) vs 40.0 (≤30 yrs) | 38.4 (1) | <0.001 | Significant |
| Education | 74.0 (no formal) vs 25.0 (tertiary) | 9.1 (3) | 0.028 | NS (after correction) |
| Location | 80.0 (rural) vs 60.0 (urban) | 14.7 (1) | <0.001 | Significant |

NS = Not Significant after Bonferroni correction.

(82.7% of males vs 49.3% of females reported self-medicating their birds). Older farmers also had higher rates: 75.3% of farmers older than 30 years practiced ASM compared to 40.0% of those aged 18–30 years. Education showed an inverse relationship – 74% of farmers with no formal education practiced ASM, versus an estimated ~25% of those with tertiary education – suggesting that more educated farmers might be somewhat less prone to misuse antibiotics, though even in secondary-educated farmers the ASM rate was 60%. Farmers in rural areas had higher ASM prevalence (80%) than those in urban areas (60%). These patterns underline that male, older, less educated, and rural farmers are more likely to misuse antibiotics in poultry.

To determine independent predictors, a multivariate logistic regression including gender, age group, education, and location (Table 7) was performed. Results from multivariate logistic regression are shown as odds ratios (OR) with 95% confidence intervals in Table 7. The regression results indicate that gender, age, and location remained significant independent predictors of self-medication when controlling for the other factors, whereas the effect of education level was attenuated and not statistically significant after adjustment.

Specifically, the odds of practicing antimicrobial self-medication were approximately 5 times higher in male farmers compared to female farmers (OR ≈ 4.9, 95% CI 3.1–7.7; p < 0.001). Farmers over 30 years of age had around 4.6 times higher odds of self-medicating than those 30 or younger (OR ≈ 4.6, 95% CI 2.6–8.1; p < 0.001). Farming in a rural location doubled the odds of self-medication compared to an urban location (OR ≈ 2.7, 95% CI 1.6–4.5; p < 0.001). In the adjusted model, lower education was associated with higher odds of ASM, but this did not reach statistical significance (OR ≈ 1.3 for no formal education vs at least primary education, 95% CI 0.9–2.1; p = 0.12). This suggests that education's effect overlaps with other factors; for instance, many rural older farmers also have lower education, so the location and age factors captured much of the risk.

**Table 7. Multivariate logistic regression of factors associated with antimicrobial self-medication among poultry farmers (N = 400).**

| Factor | Comparison (Reference) | OR (95% CI | P-value |
|---|---|---|---|
| Gender | Male vs Female | 4.9 (3.1–7.7) | < 0.001 * |
| Age | >30 years vs ≤ 30 years | 4.6 (2.6–8.1) | < 0.001 * |
| Education | No formal vs ≥Primary | 1.3 (0.9–2.1) | 0.12 |
| Location | Rural vs Urban | 2.7 (1.6–4.5) | < 0.001 * |

*p < 0.001 (statistically significant).

**Model diagnostics.** Hosmer–Lemeshow test: χ² = 15.51, df = 8, p = 0.030. While the model fits reasonably, some miscalibration across deciles was observed (S3 Table).

ROC analysis: The ROC curve showed good discrimination, with AUC = 0.77. (i.e., it can distinguish between farmers who self-medicate and those who do not with ~77% accuracy) as shown in Fig 2.

## Discussion

### Socio-demographic characteristics of poultry farmers and factors associated with antimicrobial self-medication

This study provides a comprehensive look at poultry farmers' antimicrobial use practices and awareness of AMR in Accra, Ghana. The findings reveal a troubling pattern of frequent antibiotic misuse and low awareness of antimicrobial resistance, which has serious implications for animal and public health. The discussions focus on key findings, compare them with other studies, and consider their significance in the context of AMR containment efforts.

Older farmers were significantly more likely to self-medicate their poultry with antimicrobials than younger farmers (75.3% vs 40% practicing ASM). One possible reason is that older poultry farmers rely on established routines and personal experience. Having been in the business longer, they may have historically used antibiotics as preventive or routine measures and may be less inclined to change those practices [8]. They might also have witnessed disease

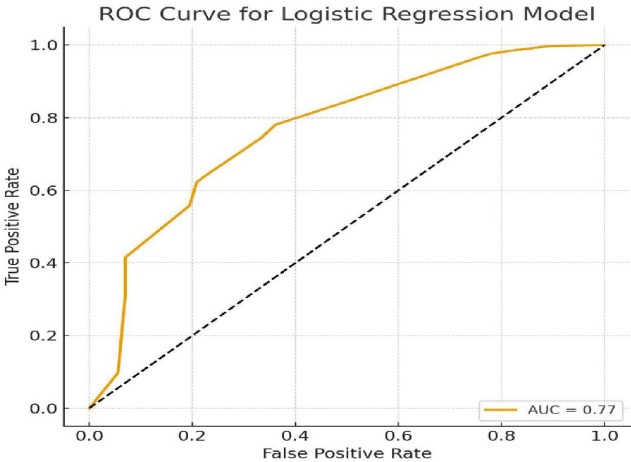

The ROC curve showed good discrimination, with AUC = 0.77, distinguishing between farmers who self-medicate and those who do not with ~77% accuracy

**Fig 2. ROC curve for logistic regression model.** Multicollinearity check: Variance Inflation Factors (VIFs) were < 2 for all predictors, indicating no collinearity issues.

outbreaks and developed a habit of aggressive antibiotic use as a safeguard. In contrast, younger farmers might have had more recent exposure to education on judicious antibiotic use or awareness of AMR through media or training programs [9,10]. Younger individuals are often more adaptable to new information and technologies – for instance, they may use smartphones or online resources to learn about poultry health or be part of social networks where responsible AMU is discussed. Younger farmers could thus be more receptive to implementing biosecurity measures or seeking veterinary advice, rather than automatically reaching for antibiotics. Additionally, peer influence differs by age: older farmers might reinforce each other's traditional practices in their networks, whereas younger farmers might form groups that discuss modern farming practices, including limiting antibiotic use. Nonetheless, it is important to note that not all older farmers misuse antibiotics – some are open to change – and not all younger farmers use them wisely, but the trend suggests age-tailored approaches in education campaigns may be beneficial [4].

The results indicated that male farmers (82.7%) misused antimicrobials at a much higher rate than female farmers (49.3%). This likely reflects gender roles in farming within the region. Men often control farm management decisions and finances, including purchasing and administering drugs, whereas women's roles might centre on day-to-day animal care (feeding, cleaning) rather than medicine administration [11]. Women in these communities also tend to have less access to formal agricultural training and veterinary information due to socio-cultural factors and lower literacy on average [12]. This knowledge gap might ironically result in women being *less* likely to use antibiotics unless instructed, whereas men, confident in their knowledge (whether accurate or not), may freely use antibiotics as they see fit. Moreover, decision-making power often lies with men in both household and commercial settings; if women farmers lack authority to seek veterinary care or spend money on professional services, they might simply follow the limited options available, which could mean either not treating or using fewer drugs [13]. On the other hand, male farmers may have greater social networks and interactions (e.g., at markets, farmer associations) where information – or misinformation – about antibiotics circulates [8,14]. They might share tips like using certain antibiotics as growth promoters, thereby normalizing misuse. Gender-targeted interventions could be useful: empowering female farmers with knowledge and decision-making capacity and engaging male farmers through peer champions could both improve antimicrobial stewardship. Additionally, gender-specific training programs should be considered, as recommended by other studies, to address unique challenges faced by women (like access to credit or information) and encourage responsible use among men [4].

Education level showed a clear gradient in bivariate analysis, with less-educated farmers practicing more antibiotic misuse. Those with tertiary education comprised only 12% of the sample but appeared to use antibiotics more prudently (an estimated 25% ASM rate among tertiary-educated, versus ~74% in illiterate farmers). Education likely enhances one's capacity for critical thinking and understanding of scientific concepts [5]. Farmers with higher education are more likely to grasp how AMR develops and the importance of responsible antibiotic use. They might also be better informed about alternative disease prevention methods, such as vaccination, probiotics, or improved sanitation, and more willing to adopt those practices [15]. Educated farmers can read and understand drug labels or published guidelines, enabling them to follow dosage instructions and withdrawal periods. They are also typically more aware of regulations and more inclined to comply with them [16]. Farmers without formal education may rely on anecdotal knowledge or advice from drug sellers that could be inaccurate. They may not appreciate the long-term consequences of misuse, focusing instead on short-term benefits like quick disease fixes. This underscores that farmer education programs are crucial. Indeed, providing even basic training on antibiotic use and resistance to farmers with minimal formal education can significantly change practices [17]. Encouragingly, the findings on attitude show farmers strongly agree that public awareness is needed – reflecting a demand for more knowledge. Expanding agricultural extension services to educate farmers at all literacy levels about AMR and promoting simple guidelines (using pictograms or local language materials for those who cannot read English) could improve understanding. Over time, as education improves, there will be more rational antibiotic use, but in the interim, targeted awareness campaigns in local communities are needed.

The study included an equal split of urban and rural farmers and observed that rural farmers were more likely to misuse antibiotics (80% ASM) than urban farmers (60%). Rural areas generally have limited access to veterinary services – for example, there may be few, or no veterinary stationed in remote communities, whereas urban Accra has more private veterinary services and agro-pharmacies. Rural farmers, facing livestock illnesses, might have no choice but to use whatever drugs they can obtain, often without proper guidance [18]. They may also perceive veterinary consultation as costly or logistically difficult, prompting self-reliance. In contrast, urban farmers, being closer to services, might occasionally utilize their services or at least have better exposure to veterinary advice when buying drugs. Furthermore, differences in regulatory enforcement could play a role: urban pharmacies might slightly better enforce prescription requirements (though still weakly), while rural shops often sell any antibiotics over-the-counter unchecked. Additionally, disease pressures can differ – some rural farmers indicated that disease outbreaks and poor biosecurity led them to routinely use antibiotics as a preventive crutch. In urban settings, farmers might be more conscious of consumer perceptions and market demands which can motivate more careful use [19]. The finding that geography influences AMU means interventions should ensure rural coverage, perhaps through mobile veterinary clinics or training community animal health workers to reach remote farmers [6]. Strengthening veterinary infrastructure in rural areas and subsidizing veterinary care for small-scale farmers could reduce the need for self-medication.

## Antimicrobial resistance awareness and knowledge

The knowledge, attitudes and practice (KAP) results together depict a situation where knowledge is lagging, attitudes are cautiously positive, and practices are problematic. Most farmers did not know about AMR or did not connect their antibiotic use with resistance problems. This aligns with other KAP surveys in African livestock sectors that have found limited awareness of AMR among farmers [15,20]. Despite this, many farmers in this study agreed in principle that antibiotics should be used carefully – for example, an overwhelming majority said professional advice is important and that regulations and awareness could help. This reflects a potentially receptive audience for behaviour change: farmers are not necessarily opposed to prudent use; rather, they lack practical knowledge and support to implement it. The mismatch between attitudes and practice (e.g., 76% agree vets should be consulted, yet 70% self-medicate without vets) often comes down to structural and economic constraints. Farmers might value veterinarians' role but find it impractical or too expensive to seek their services, especially for smaller operations. Similarly, while they conceptually support regulations, they might fear that strict enforcement could limit their access to life-saving drugs for their flocks. Therefore, efforts to change practices must go hand in hand with making veterinary care more accessible and building trust.

## Antimicrobial usage practices

The findings on antibiotic usage are also consistent with research from other regions of Ghana and Africa. For instance, Boamah et al. (2016) reported extensive tetracycline and sulfonamide use in Ghanaian poultry farms [20], and similar patterns were noted in Nigeria [18]. Such classes are inexpensive and readily available, explaining their popularity. However, bacteria like *E. coli* and *Salmonella* isolated from poultry in Ghana have shown high resistance to these commonly used antibiotics [21–22]. A recent study in Uganda found over 50% of *E. coli* from small-scale poultry farms were multidrug-resistant [8]. In this study area, it is likely that resistance is already widespread: although microbiological testing was not performed, other research in Accra's rural farms has isolated *coagulase-negative staphylococci*, *Salmonella*, and *Clostridium* species from poultry that are often resistant to multiple drugs [21–23]. Moreover, multi-drug resistance (MDR) rates reported in literature are alarming – for example, 83.3% of *Salmonella* species and 80.0% of *Staphylococcus* species from some poultry studies were MDR [24]. These statistics underline the potential threat: if farmers continue current practices, infections in poultry may become increasingly difficult to treat with standard antibiotics.

The unregulated use of antibiotics in Accra's poultry sector, as evidenced by this study, has several implications. Firstly, it facilitates the emergence of resistant zoonotic pathogens. Poultry can act as reservoirs for resistant *E. coli*, *Salmonella*,

and *Campylobacter* strains that can infect humans. For example, poultry-associated *Salmonella* outbreaks can occur, and if the strain is drug-resistant due to farm antibiotic use, treating human cases becomes challenging. Secondly, drug residues in poultry meat or eggs could pose health risks and contribute to resistance if consumers are consistently exposed to low antibiotic doses [18]. The widespread use of antibiotics like tetracyclines and nitrofurans might lead to residues in food products; if withdrawal periods are not observed (and our findings suggest many farmers are not aware of or not following such guidelines), consumers could ingest these residues unknowingly [6]. Nitrofurans (e.g., furazolidone) are antimicrobial drugs often used in poultry for bacterial enteritis; their high usage is notable given that some nitrofurans are banned in food animals in many countries due to carcinogenic residue concerns. Nitrofurans persist because they are accessible, cheap, and enforcement is weak; but their carcinogenic residues and AMR selection make them high-risk for public health, food safety, and the environment. The current data mirror broader Ghana/West Africa patterns and underscore the need for enforceable bans, residue testing, and practical farm-level alternatives within a One Health program [25]

Thirdly, in the animal health context, overuse of antibiotics can mask underlying management problems – farmers might rely on drugs instead of improving hygiene and vaccination protocols, which is not sustainable. Eventually, if antibiotics lose efficacy due to resistance, farmers could face devastating losses from disease outbreaks that no longer respond to treatment.

These findings mirror trends observed in other studies of poultry farms in West Africa, which report heavy usage of tetracyclines and other broad-spectrum antibiotics [20]. The presence of colistin use, albeit in only 1% of farmers, is noteworthy because colistin is considered a last-resort antibiotic in human medicine for certain multidrug-resistant infections. Its use in animals is highly regulated in some countries, but here a few farmers reported using it, likely as a gut health promotor or treatment for severe infections. The diversity of antibiotics being used – including critically important drugs – underscores the risk of selecting for resistant pathogens like *E. coli*, *Salmonella*, and *Staphylococcus* on these farms. In fact, prior studies in Ghana and neighbouring countries have documented high levels of resistance in poultry bacterial isolates to tetracyclines, penicillins, and fluoroquinolones, correlating with their extensive use [20–22]. Farmers' choices of antibiotics are often guided by availability and cost, as well as informal advice, rather than laboratory diagnosis or sensitivity testing. This pattern of usage highlights the urgent need for antibiotic stewardship and regulation in the poultry sector to prevent the propagation of AMR.

A strength of this study is the large sample size (N = 400) and inclusion of diverse communities, enhancing the representativeness of the findings for the Accra area poultry farming population. The use of systematic random sampling and researcher-administered questionnaires likely improved the accuracy of responses. The comprehensiveness of the questionnaire, covering knowledge, attitudes, and practices, allows correlating these aspects and get a fuller picture of farmer behaviour. The use of statistical analysis to rigorously identify factors associated with misuse, providing evidence for targeted interventions.

However, some limitations must be noted. The study is cross-sectional, so it captures practices at one point in time, which cannot establish causality or how practices might change over seasons or years. Another limitation is the reliance on self-reported data, which may be subject to recall bias or social desirability bias. This was mitigated by showing pictures of common drug packages and asking respondents to identify what they use, but errors are possible. Resistance risk was inferred from usage patterns and external data rather than demonstrating it directly on these farms – integrating microbiological surveillance would strengthen the link between use and resistance outcomes.

## Conclusions

The current study has demonstrated that the use of antibiotics in poultry farming in the Greater Accra Region of Ghana is widespread and often unregulated. Farmers' priorities – preventing and treating diseases to protect their livelihoods – coupled with the easy availability of antibiotics without veterinary prescription have led to prevalent practices of antibiotic self-medication in poultry. Many farmers, particularly males and those with more experience, administer important

antibiotics (including tetracyclines, penicillins, and fluoroquinolones) with minimal adherence to recommended usage guidelines or withdrawal periods, with or without professional guidance. This indiscriminate use of antimicrobials is contributing to the silent development of antimicrobial resistance on farms, which poses a threat to animal health, public health, and food safety.

The findings underscore that AMR is not yet a well-understood concept among poultry farmers – a majority were unaware of resistance development or did not connect their antibiotic use to potential treatment failures in the future. However, the farmers' generally positive attitudes toward seeking advice and supporting regulations provide a hopeful avenue for change. There is an urgent need for public health interventions and stricter regulatory enforcement to promote the prudent use of antibiotics in food-producing animals. Such measures will protect the efficacy of critical antibiotics for both animal and human medicine. The problem of AMR in agriculture spans across sectors, reinforcing the importance of a One Health approach: coordinated action that involves veterinary services, public health officials, farmers, and environmental agencies. By working together to improve communication, education, and oversight, stakeholders can raise awareness of AMR and encourage behaviour change. Only through multi-sectoral collaboration and commitment can we ensure that antibiotic use in poultry farming is optimized, to preserve the effectiveness of antimicrobial drugs for future generations.

## Supporting information

**S1 File. Questionnaire with informed consent form.**
(DOCX)

**S2 Data. Anonymized Data.**
(DOCX)

**S3 Table. Hosmer-Lemeshow test table.**
(XLSX)

## Acknowledgments

The authors thank all poultry farmers who participated in this study for their time and cooperation.

## Author contributions

**Conceptualization:** Henry Kwadwo Hackman.

**Data curation:** Henry Kwadwo Hackman, Lawrence Annison, Reuben Essel Arhin, Alice Constance Mensah, Evans Andoh Wilberforce, Louis Appiah, Sharon Annison.

**Formal analysis:** Henry Kwadwo Hackman, Alice Constance Mensah, Ebenezer Krampah Aidoo, Evans Andoh Wilberforce, Louis Appiah, Sharon Annison, Matilda Ayim Akonor.

**Funding acquisition:** Henry Kwadwo Hackman, Lawrence Annison, Reuben Essel Arhin, Alice Constance Mensah, Ebenezer Krampah Aidoo, Matilda Ayim Akonor.

**Investigation:** Henry Kwadwo Hackman, Evans Andoh Wilberforce.

**Methodology:** Henry Kwadwo Hackman, Reuben Essel Arhin, Ebenezer Krampah Aidoo, Evans Andoh Wilberforce.

**Project administration:** Henry Kwadwo Hackman, Lawrence Annison.

**Resources:** Henry Kwadwo Hackman.

**Supervision:** Henry Kwadwo Hackman.

**Validation:** Henry Kwadwo Hackman, Lawrence Annison, Reuben Essel Arhin, Alice Constance Mensah, Ebenezer Krampah Aidoo, Louis Appiah, Sharon Annison, Matilda Ayim Akonor.

**Visualization:** Henry Kwadwo Hackman, Reuben Essel Arhin.

**Writing – original draft:** Henry Kwadwo Hackman, Evans Andoh Wilberforce.

**Writing – review & editing:** Henry Kwadwo Hackman, Lawrence Annison, Reuben Essel Arhin, Alice Constance Mensah, Ebenezer Krampah Aidoo, Sharon Annison, Matilda Ayim Akonor.

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
