## [Decision Letter · Decision Letter 0]

12 Sep 2025

Dear Dr. Hackman,

We look forward to receiving your revised manuscript.

Kind regards,

Anselme Shyaka, Ph.D

Academic Editor

PLOS ONE

Journal Requirements:

2. In the ethics statement in the Methods, you have specified that verbal consent was obtained. Please provide additional details regarding how this consent was documented and witnessed, and state whether this was approved by the IRB.

Additional Editor Comments :

Thanks for submitting your manuscript for consideration. Before we can proceed toward publication, please revise the work as follows (in addition to addressing reviewer comments):

1. Introduction – Focus & Knowledge Gap

Make the introduction concise, avoid a broad AMR review, and focus on the Ghanaian context. Clearly state the gap and what new insight this study provides compared to previous Ghana/West Africa studies.

2. Methods – Sampling & Sample Size

Clarify how farmer lists were obtained, how random selection was done, and the response rate. Explain the adjustment from the Cochran formula (~384) to 400 respondents.

3. Statistical Analysis – Chi-square & Model Fit

Address the risk of inflated Type I error from multiple chi-square tests (e.g., adjustment or limitation). Include model diagnostics (e.g., Hosmer–Lemeshow test) to confirm logistic regression fit.

4. Results – Internal Consistency

Check and reconcile numbers (e.g., Table 1 age percentages, attitude responses >100%). Align tables with narrative.

5. Discussion – Depth & One Health

Move beyond repeating results. Explain why nitrofurans are still used, discuss implications for zoonoses, food safety, and environment, compare with local/regional studies, and give actionable recommendations.

6. Ethics Statement

Resolve the inconsistency between the submission form (“N/A”) and cited ethics approval.

7. Data Availability & Transparency

Deposit anonymized data in a public repository and share the DOI/link. Include questionnaire and scoring rubric as supplementary files for reproducibility.

Reviewers' comments:

Reviewer's Responses to Questions

**Comments to the Author**

1. Is the manuscript technically sound, and do the data support the conclusions?

Reviewer #1: Yes

Reviewer #2: Yes

2. Has the statistical analysis been performed appropriately and rigorously?

Reviewer #1: Yes

Reviewer #2: Yes

3. Have the authors made all data underlying the findings in their manuscript fully available?

Reviewer #1: No

Reviewer #2: Yes

4. Is the manuscript presented in an intelligible fashion and written in standard English?

Reviewer #1: Yes

Reviewer #2: Yes

Reviewer #1: This article presents the results of a cross-sectional survey conducted to assess knowledge, attitudes, and practices (KAP) regarding antimicrobial use (AMU) and antimicrobial resistance (AMR) among 400 farmers in Accra, Ghana. It is an interesting study that aims to fill existing information gaps on AMU/AMR in Ghana.

Although the manuscript is generally well written, the Methodology section needs improvement. It should provide more detail on the data collection process and the criteria used to classify farmers into different categories. Additionally, the citations are not well organized; the authors should revise the in-text citations so that references appear in ascending numerical order.

The data analysis could also be improved to present the findings in a more comprehensive and reader-friendly manner. Finally, the Discussion section should be restructured to follow the same sequence as the Results section for better coherence.

Please find additional comments in the attached document.

Reviewer #2: General comment

The authors present the results of the original research on the awareness of poultry farmers on the prudent use of antimicrobials. Conclusions are drawn from the obtained data. The study investigated the knowledge, attitudes, and practices of poultry farmers. Inappropriate antimicrobial use is prevalent in the study area, Ghana and the awareness is low among poultry farmers. However, the positive attitudes positive attitudes toward seeking advice and supporting regulations provide a hopeful avenue for change. There is an urgent need for public health interventions and stricter regulatory enforcement to promote the prudent use of antibiotics in food-producing animals. Such measures will protect the efficacy of critical antibiotics for both animal and human medicine. The problem of AMR in agriculture spans across sectors, reinforcing the importance of a One Health approach: coordinated action that involves veterinary services, public health officials, farmers, and environmental agencies.

However, the study has some limitations including the low knowledge of names/families of antibiotics used by poultry farmers, and lack of bacteriological experiments of AMR. However, authors developed strategies to overcome the limitations including using antibiotic’s package images and reporting resistance findings of Salmonella, E. Coli, and Staphylococcus from other studies.

Areas of improvement

Line 55: in the key words section, delete Bacterial species including Escherichia coli, salmonella which were not studies for AMR in this study.

Lines 62, 65 etc. [3][4]. Combine references in one blanket [3-4] and harmonize for the whole manuscript.

Table 2. Replace the question by a positive statement for consistency. Replace “can you reduce AMR development?” by “avoiding overuse of antimicrobials in animal production can reduce AMR development”

Replace “Can your imprudent use of antimicrobials affect the health of others in the form of AMR?” by “Imprudent use of antimicrobials can affect the health people in the form of AMR”

Line 183. Delete “When asked What do you do when your animals get sick?”

Authors have included several discussion statements in the results section. Some of the same statements are repeated in the discussion section. Authors should describe their findings in the results section and move the discussion statement to the discussion section. Some examples:

Lines 188-190: These findings show that about 70% of farmers resort to treating their sick poultry without professional veterinary care, either by themselves or with peer advice, which raises concerns about inappropriate drug use,

Lines 203-206: This suggests that although veterinary involvement in actual treatment is low, about two-thirds of farmers at least speak with a veterinary or animal drug vendor to get suggestions for what drugs to buy – but one-third engage in completely unguided antibiotic purchases,

Lines 215 -217: an injudicious practice where low doses of antibiotics are added to feed to boost weight gain. Although this 4% appears as a small minority, it reflects the persistence of growth promotion uses that have been banned or discouraged in many countries due to AMR concerns.),

Lines 221-226: This indicates that while some farmers do access antibiotics through formal veterinary channels (clinics or pharmacies), an even larger share rely on informal local shops. In all cases, antibiotics are readily available without strict prescription enforcement. The ease of access to antimicrobials in the community – often without regulation – was confirmed by many respondents and is consistent with reports from other low-income settings [24]

Line 237 – the criteria for Positive attitudes should be clearly explained in method section

Lines 243-245: This level of agreement --------- should move to discussion section

Lines 252-253: The mixed of responses ……………….. should be part of discussion section

Line 271: This reflects …. Part of discussion section

Lines 277 -280: This suggests that ……………… part of the discussion section

Lines 283 – 286: The arguments are mixed and confusing. (53% agreements for education then 70%). Is the idea in line 282 the same like in 283 (specifically ……)? Rephrase to make sense If not better to start a new paragraph to sum up all attitude’s responses.

Lines 287 – 290: it is evident ……………. Part of discussion

Lines 291-298: Part of discussion section

Lines 303-308: part of discussion section

Lines 313 – 315: Part of discussion

Lines 303 -305: Antibiotics usage – looking at the education of farmers one can wonder their level of knowledge with antibiotics names

Lines 326 – 339: part of discussion

Lines 374 - 389: part of discussion

Discussion: Authors would focus their discussion to the important findings and exclude non-significant predictors of AMR of this study such as education and geography (rural and urban).

Lines 510 -516: include references to support your statements

Lines 532 – 358: authors present extensive limitations which undermine the quality of the study. They can limit limitations to absent of bacteriological AMR experiments, and knowledge of previously used antibiotics among majority of poultry farmers.

**Do you want your identity to be public for this peer review?** For information about this choice, including consent withdrawal, please see our Privacy Policy

Reviewer #1: No

Reviewer #2: **Yes: ** Jean Bosco Ntivuguruzwa

---

## [Author Response · Author response to Decision Letter 1]

13 Oct 2025

Responses to Reviewer 1 and Reviewer 2 have been indicated in the responses to reviewers' document.

---

## [Decision Letter · Decision Letter 1]

4 Nov 2025

Following the second review, some minor revisions are still required to fully address the points raised by the reviewers. We invite you to submit a revised version of your manuscript incorporating these final changes.

We look forward to receiving your revised manuscript.

Kind regards,

Anselme Shyaka, Ph.D

Academic Editor

PLOS ONE

Journal Requirements:

Reviewers' comments:

Reviewer's Responses to Questions

**Comments to the Author**

Reviewer #1: (No Response)

Reviewer #2: All comments have been addressed

2. Is the manuscript technically sound, and do the data support the conclusions?

Reviewer #1: Yes

Reviewer #2: Yes

3. Has the statistical analysis been performed appropriately and rigorously?

Reviewer #1: Yes

Reviewer #2: Yes

4. Have the authors made all data underlying the findings in their manuscript fully available?

Reviewer #1: Yes

Reviewer #2: Yes

5. Is the manuscript presented in an intelligible fashion and written in standard English?

Reviewer #1: Yes

Reviewer #2: Yes

Reviewer #1: The authors have improved the manuscript to bring more clarity. Here are some minor comments:

- Please revise table 5 : For Education the Total of farmers with ASM does not add up to 280 as for other factors.

- It seems like all respondents agreed to be or were identified as Antimicrobial users. to avoid confusion, please add this data somewhere in the manuscript (all respondents were identified as using antibiotics in their animals.

- Line 466: Rectify: antimicrobial misuse causing AMR…. Instead of antimicrobial use causing AMR….

Reviewer #2: All comments were addressed. However, I have two suggestions:

1) authors added two tables 6, and 7 which are consecutive in the manuscript. Tables 5 and 6 should be placed next to the paragraph they are cited (between 299-300).

2) The discussion section has subheadings which is not common in the discussion section.

**Do you want your identity to be public for this peer review?** For information about this choice, including consent withdrawal, please see our Privacy Policy

Reviewer #1: No

Reviewer #2: **Yes: ** Jean Bosco Ntivuguruzwa

---

## [Author Response · Author response to Decision Letter 2]

5 Nov 2025

All responses to reviewers have been included in the responses to reviewers' table.

---

## [Editor Report · Decision Letter 2]

10 Nov 2025

Awareness of antimicrobial resistance and antibiotic use among poultry farmers in Accra, Ghana: a cross-sectional survey

PONE-D-25-35681R2

Dear Dr. Hackman,

We’re pleased to inform you that your manuscript has been judged scientifically suitable for publication and will be formally accepted for publication once it meets all outstanding technical requirements.

Kind regards,

Anselme Shyaka, Ph.D

Academic Editor

PLOS ONE
---

## [Editor Report · Acceptance letter]

PONE-D-25-35681R2

PLOS ONE

Dear Dr. Hackman,

I'm pleased to inform you that your manuscript has been deemed suitable for publication in PLOS ONE. Congratulations! Your manuscript is now being handed over to our production team.

Kind regards,

on behalf of

Dr. Anselme Shyaka

Academic Editor

PLOS ONE